# A meta-analysis of birth-origin effects on reproduction in diverse captive environments

Katherine A. Farquharson[1], Carolyn J. Hogg[1] & Catherine E. Grueber [1,2]

Successfully establishing captive breeding programs is a priority across diverse industries to address food security, demand for ethical laboratory research animals, and prevent extinction. Differences in reproductive success due to birth origin may threaten the long-term sustainability of captive breeding. Our meta-analysis examining 115 effect sizes from 44 species of invertebrates, fish, birds, and mammals shows that, overall, captive-born animals have a 42% decreased odds of reproductive success in captivity compared to their wild-born counterparts. The largest effects are seen in commercial aquaculture, relative to conservation or laboratory settings, and offspring survival and offspring quality were the most sensitive traits. Although a somewhat weaker trend, reproductive success in conservation and laboratory research breeding programs is also in a negative direction for captive-born animals. Our study provides the foundation for future investigation of non-genetic and genetic drivers of change in captivity, and reveals areas for the urgent improvement of captive breeding.

[1] The University of Sydney, School of Life and Environmental Sciences, Faculty of Science, Sydney NSW 2006, Australia. [2] San Diego Zoo Global, PO Box 120551 San Diego, CA 92112, USA. Correspondence and requests for materials should be addressed to C.E.G. (email: catherine.grueber@sydney.edu.au)

Animals have been kept by humans since the change from a hunter-gatherer lifestyle to farming ~8500 years ago[1]. Successful reproduction is the most fundamental requirement of captive breeding programs across a range of industries including commercial production, conservation, and research. The domestication of wild animals involves increasingly diverse species to address global food security[2]. In particular, the growth of the aquaculture industry from less than one million tons of aquatic food (including fish, crustaceans, molluscs, echinoderms, and amphibians) in the 1950s, to an expected 85 million tons by 2030, has driven the diversification of species used[2]. The establishment of closed-cycle breeding programs is essential for the growth and sustainability of aquaculture as wild fish-stocks continue to be depleted[3]. In conservation, captive breeding has been recommended by International Union for the Conservation of Nature (IUCN) Red List assessors for 2199 species as a tool to reduce the threat of extinction[4]. For research populations, some countries have banned the use of wild-caught non-human primates in modern laboratory research and insufficient captive-born animals are produced to meet demand[5]. Successful captive breeding, as opposed to continual supplementation of captive populations with wild animals, can also help avoid additional welfare concerns arising from wild-born animals adjusting to a captive environment[6]. Thus, identifying limitations

**Fig. 1** Phylogenetic tree of 44 species included in the meta-analyses. The tree was created using the 'rotl' package[58] in R. The total number of comparisons between captive-born and wild-born animals included for each species is given as (N)

**Table 1 Meta-analytic effect size estimates of differences in reproductive success between wild-born and captive-born animals in captive environments**

| | Posterior mode (lnOR) [95% HPD CI] | % odds of captive-born reproductive success | % odds of wild-born reproductive success | N |
|---|---|---|---|---|
| Overall model* | −0.56 [−1.01, −0.10] | −42.3% | +74.2% | 115 |
| Overall model + phylogeny | −0.65 [−1.45, 0.04] | −47.7% | +91.3% | 115 |
| *Captive environment* | | | | |
| Aquaculture* | −1.45 [−2.46, −0.56] | −76.7% | +328.7% | 23 |
| Conservation | −0.38 [−1.06, 0.30] | −31.8% | +46.6% | 51 |
| Research | −0.34 [−1.08, 0.35] | −29.0% | +40.8% | 40 |
| Other | 1.84 [−0.98, 4.49] | +527.6% | −84.1% | 1 |
| *Trait type* | | | | |
| Fertility and hatchability | −0.38 [−0.94, 0.15] | −31.5% | +45.9% | 30 |
| Reproductive yield | −0.52 [−1.06, 0.05] | −40.6% | +68.4% | 28 |
| Offspring quality* | −1.22 [−2.01, −0.46] | −70.5% | +238.8% | 8 |
| Offspring survival* | −1.26 [−1.85, −0.65] | −71.5% | +250.9% | 33 |
| Reproductive phenology | −0.04 [−0.69, 0.57] | −3.5% | +3.6% | 16 |

Posterior mode gives the meta-analytic log odds ratio (lnOR) estimate from the MCMCglmm models, with lower and upper 95% higher posterior density credible intervals given. Estimates with the 95% HPD CI excluding zero are marked with *. Percentage odds refers to the % increase (+) or decrease (−) in the odds of reproductive success of captive-born or wild-born animals, relative to the other group

and opportunities for captive breeding across all industries is an urgent priority.

Considerable research has explored differences between captive and wild populations in terms of their health, genetics, nutrition, behavior, physiology, and reproduction (for examples see refs. [7–13]). However, far less attention has been given to differences that may exist between wild-born and captive-born animals when both are considered in a captive environment. Although many breeding programs aim to replicate some wild conditions in the captive environment in order to promote successful reproduction, it is inevitable that differences in nutrition, social structures, and breeding strategies will occur. Genetic change in captive populations is likely, and potentially unavoidable, as a result of founder effects, inbreeding, drift and adaptation to captivity, among other processes[14]. If these processes combine to result in captive-born animals that are less successful than their wild-born counterparts, closed-cycle aquaculture may not be economically viable and the long-term sustainability of conservation breeding and laboratory research is threatened[15]. Conversely, genetic adaptation to captivity may increase the reproductive success of captive-born animals, however this comes at the cost of a potential reduction in fitness if animals are released to the wild[14,16].

Genetic change in captivity may be beneficial or deleterious depending on a program's goals. Aquaculture systems aim to domesticate species through selecting highly productive individuals over generations of captive breeding[17], while conservation breeding programs aim to avoid selection[18] in order to retain wild traits and genetic diversity in the eventual prospect of reintroduction to the wild[19]. The role of selection in research breeding programs is less clear and depends on the species involved and the purpose of the research. Nevertheless, all three of these captive breeding industries share a reliance on successful reproduction among captive-born animals. Differences in reproductive success as a result of birth origin may arise as a result of genetic effects such as inbreeding depression[20] and adaptation to captivity[21]; non-genetic effects, such as inappropriate social development, stress[22], and nutrition[23]; and complex interactions, such as the early rearing environment and maternal effects[24]. Due to this complexity, assessing the success of captive breeding programs by examining only one metric, such as breeding success (i.e., producing an offspring), fails to account for life-history trade-offs that may occur, and/or differential impacts of captivity throughout a species' life history. For example, if captive-born animals produce more offspring per breeding event than their wild-born counterparts but have higher juvenile mortality, lifetime reproductive success (i.e., total genetic contribution to the next generation), may be similar to, or perhaps even lower, than wild-born individuals.

Birth-origin effects have been examined in a number of species with mixed results[22,25]. As the majority of studies in this area focus on single species, it has not previously been possible to quantitatively ascertain whether differences in reproductive success follow general trends across taxa and captive environments or whether they are specific to the study species, the captive environment of interest, or the type of reproductive trait examined. Here we provide a systematic review and meta-analysis to examine the influence of birth origin on reproductive success across multiple species, a variety of life-history traits and in various captive environments. We take a broad definition of 'reproductive success' to refer to diverse measures of reproductive traits, encompassing production of gametes/offspring at multiple stages throughout the life history of breeders. Specifically, our objective was to quantify differences in reproductive success between captive-born and wild-born animals, in captivity, across diverse animal species to determine whether birth-origin effects are specific to taxa or follow a general trend regardless of phylogeny. As all captive breeding programs (aquaculture, conservation, and laboratory research) require successful reproduction for their management objectives, all are included in this review. Diverse literature (115 effect sizes from 44 species) shows that, overall, captive-born animals have a 42% decreased odds of reproductive success in captivity compared to their wild-born counterparts. The strongest trends are seen in commercial aquaculture settings, with weaker effects (but in the same direction), in conservation and laboratory settings. The choice of traits measured also impacts the reported effect of birth origin on reproductive success, with offspring survival and quality being the most sensitive traits. Examining varied measures of reproductive success in this study gives insight into the possible drivers of birth-origin effects that have important implications for the establishment, efficacy and long-term viability of captive breeding programs.

## Results

**Wild-born animals are more productive in captivity.** A total of 39 papers published between 1967 and 2015 contributed 115

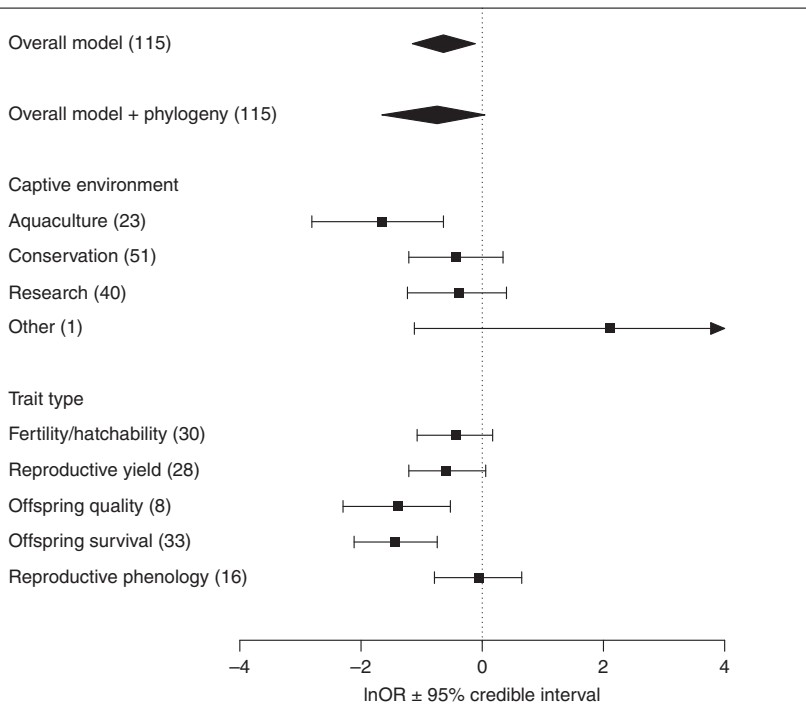

**Fig. 2** Forest plot of overall meta-analytic results (diamonds), and meta-regression models of captive environment and trait type (squares). A negative log odds ratio (lnOR) indicates wild-born animals have higher reproductive success than their captive-born counterparts, with a positive log odds ratio referring to increased reproductive success of captive-born animals compared to wild-born. Squares represent the posterior mode (or parameter estimate) with error bars showing the 95% highest posterior density credible intervals (95% HPD CIs). *N* refers to the number of effect sizes. See Methods section for definition of study environments, and Supplementary Table 2 for the comparisons included in each trait type category

comparisons of reproductive traits between captive-born and wild-born animals in captive environments for analysis (some papers compared more than one reproductive measure, or more than one species) (Supplementary Data 1). The final dataset included 44 species from phylogenetically diverse taxa including invertebrates, fish, birds, marsupials, and eutherian mammals (Fig. 1). We used the log odds ratio to quantify the standardized effect size of differences in reproductive success, where negative log odds ratios represent higher reproductive success of wild-born animals compared to their captive-born counterparts. Overall, wild-born animals have a 74.2% (lnOR = −0.56, 95% HPD CI: [−1.01, −0.10], Table 1) increased odds of reproductive success in captive environments compared to captive-born animals, equivalent to a small-medium effect size. Accounting for phylogenetic non-independence occurring as a result of shared evolutionary history did not greatly alter the point estimate but broadened the CI resulting in it crossing zero (Fig. 2). Phylogeny accounted for only 0.29% of heterogeneity (see Supplementary Table 1 for full extended heterogeneity statistics for both models), and phylogenetic heritability was low ($H^2 = 0.0026$), therefore our result is generalizable across species. As the non-phylogenetic model had a lower DIC than the phylogenetic model (DIC = 317.4 vs. 317.5) all subsequent analysis (meta-regression) proceeded without phylogeny. The high heterogeneity ($I^2_{total} = 94\%$) observed within our dataset is not surprising given the diverse species, captive environments and reproductive traits included, and is consistent with other ecological and evolutionary meta-analyses[26]. We next examined the source of this heterogeneity by fitting moderator variables to our analysis, and determined the contribution of environment and trait type on the effect of birth origin.

**Birth-origin effects vary with captive environment**. Our dataset included data collected from four study environments: aquaculture (N = 9 publications, 23 comparisons), conservation (N = 

14 publications, 51 comparisons), research (N = 15 publications, 40 comparisons) and other (N = 1 publication, 1 comparison). Effect sizes varied according to the captive environment of the study (Fig. 2). In aquaculture systems, wild-born animals had a 328.7% increased odds of reproductive success relative to captive-born animals (lnOR = −1.45, 95% HPD CI: [−2.46, −0.56]; a large, statistically significant effect). In conservation and research environments, the estimated effect was in the same direction (negative, i.e., wild-born animals more reproductively successful), but not statistically significant at $\alpha = 0.05$ (Table 1). The one study that we categorized as 'other' examined studbook data from Burmese timber elephants[27], which are bred as working elephants and do not fit any of our other categories. The estimated effect was positive (captive-born animals were more successful), though this had poor precision (Fig. 2).

**Captive-born animals are less productive across life stages**. Our dataset included comparisons from five, broad, trait type categories: fertility/hatchability (e.g., probability of breeding; N = 30 comparisons), reproductive yield (e.g. litter size; N = 28), offspring quality (e.g., birth weight; N = 8), offspring survival (N = 33) and reproductive phenology (e.g., interbirth interval; N = 16). For a full list of the specific reproductive traits included in each category see Supplementary Table 2. Birth-origin effects were negative across all trait type categories (Fig. 2). Wild-born animals had a statistically significant 238.8% greater odds of reproductive success relative to captive-born breeders when measured as offspring quality traits, and a 250.9% greater odds of offspring survival (both considered large effects, Table 1). No statistically significant effects of birth origin were observed when reproductive success was measured as fertility/hatchability, reproductive yield or reproductive phenology (Table 1).

**Table 2 Number of effect sizes in analysis, grouped by captive environment, and trait type**

|  | Fertility and hatchability | Reproductive yield | Offspring quality | Offspring survival | Reproductive phenology | Total |
|---|---|---|---|---|---|---|
| Aquaculture | 4 | 7 | 1 | 4 | 7 | 23 |
| Conservation | 18 | 5 | 5 | 17 | 6 | 51 |
| Research | 8 | 16 | 2 | 12 | 2 | 40 |
| Other | 0 | 0 | 0 | 0 | 1 | 1 |
| Total | 30 | 28 | 8 | 33 | 16 | 115 |

**Drivers of birth-origin effects**. As the difference between wild-born and captive-born reproductive success was detectable with some trait types and not with others, we examined whether the dataset was evenly distributed across study environments and trait types (Table 2). The two trait type categories showing a strong, significant influence of birth origin, offspring quality and offspring survival, were dominated by conservation and research comparisons ($N = 7/8$, $N = 29/33$, respectively). Data from aquaculture studies contributed to all trait type categories; but data from conservation environments largely contributed to fertility/hatchability (60%), offspring quality (62.5%) and offspring survival (51.5%) (Table 2). Comparisons made in research environments comprised the largest proportion of the reproductive yield trait category (57.1%), with correspondingly high contributions to data in the fertility/hatchability and offspring survival categories.

While we were primarily interested in comparing the effect of birth origin on reproductive success, we also recorded whether each study specified the number of generations of captive breeding of the captive-born population. The majority of the studies included did not specify the generation, nor range of generations, of the captive-born population (26/39 studies, 66.7%; 80/115 comparisons, 69.6%, Supplementary Table 3), so we could not statistically analyze the effect of generations in captivity on reproductive success. Of the studies that did report generation, the most common comparison was to an F1 (first generation) captive-born population (5/13 studies, 38.5%; 17/35 comparisons, 48.6%), and this comparison was found exclusively in aquaculture and research study environments.

We found no strong evidence that our results are influenced by publication bias (Supplementary Note 1, Supplementary Fig. 1). A total of 74 comparisons of interest were excluded from our analysis because they did not report all the data required for inclusion; values for 17–41 of these comparisons could be recovered using multiple imputation (Supplementary Note 2; see Supplementary Data 1). Although multiple imputation increased uncertainty in our results, the main effects were in the same direction and of similar magnitude to those obtained by our main analysis. We therefore do not believe that our overall conclusions are biased by missing data (Supplementary Note 2, Supplementary Table 4).

## Discussion

We synthesized the results of studies across different species, captive environments, and measures of reproductive success to provide an overall estimate of the effect of birth origin on reproductive success in captivity. Our analysis included 44 species across diverse animal taxa, including vertebrates and invertebrates. Surprisingly, across all species and captive environments, it was wild-born animals that had higher odds of productivity in captivity, relative to their captive-born counterparts (74.2%). As phylogenetic signal was low (<1%), it is likely that our overall result is generalizable across species, indicating a general trend toward declines in the reproductive success of captive-born animals relative to wild-born animals, in captive environments. Our meta-analysis enables us to examine the available data on this topic in more detail, and draw inferences about the possible causes of this unexpected pattern.

When our data were stratified by captive environment, aquaculture was the only environment to show a large, statistically significant mean difference between wild- and captive-born animals (Fig. 2). Again, it was the wild-born animals that showed higher odds of reproductive success (328.7%), relative to their captive-born counterparts. This result is unexpected given that aquaculture aims to improve reproduction among captive-born stock in the process of domestication. This strong, significant effect has important implications for the sustainability of closed-cycle commercial production systems, suggesting that wild-stock supplementation, or other solutions, may be required. Effects in conservation and laboratory breeding contexts were in the same direction as seen for aquaculture, but weaker (Fig. 2). There are several possible explanations for this variation among contexts, which is consistent with the differing goals of various captive breeding programs. For example, conservation breeding programs aim to minimize adaptation to captivity[18], while phenotypic and/or marker-assisted selection for favored traits is often an important goal of agricultural breeding programs[28]. These different goals predict decreased genetic change in conservation programs, compared to agricultural programs.

A further possible source of the variation among environments is differences in the length of the captive breeding programs included in publications. Captive-born animals in different environments varied in the number of generations of captive breeding, data that were often not reported (Supplementary Table 3). For aquaculture data, 12 of the 23 comparisons originated from studies comparing the reproductive success of wild-born (F0) to first generation (F1) captive-born animals (Supplementary Table 3). It is therefore likely that much of the difference in reproductive success we observe in aquaculture is related to changes occurring within the first generation of captive breeding, rather than across multiple generations. In contrast, none of the 51 comparisons made in conservation studies specified a wild versus F1 comparison. In our conservation dataset, four comparisons were from studies comparing wild-born to a captive-born population ranging from F1–F3 or to F4, while the other 47 comparisons did not report the captive generation of comparison used in the study (Supplementary Table 3). Given the long-running nature of many conservation breeding programs, it is probable that many of the captive-born populations in those studies that did not specify the generation depth comprised a range of generations. Similarly, for research studies, captive-born animals were F1 for only 5/40 of the comparisons made, 25 of the remainder were from studies that did not report generation depth (Supplementary Table 3). As a result, we are unable to conclude whether the differences in birth-origin effects on reproductive success are influenced by general factors associated with captivity (such as purely environmental factors), or by characteristics of the captive-born population (such as genetic factors correlating with generation depth, including neutral or adaptive change). Conservation and research breeding programs could still be experiencing a reduction in reproductive success in the first generation of captive breeding. It is imperative that potential

declines in the early stages of conservation breeding programs are reported and prevented, otherwise founder genetic diversity and the evolutionary potential of the captive population may be lost[29].

Understanding possible causes of differences in reproductive success in the first generation of captive breeding is useful for the successful establishment of breeding programs. We suggest comparing changes within the first generation of captive breeding to long-term changes over multiple generations in order to disentangle possible causative factors, such as environmental effects versus long-term genetic change. For example, one possible explanation of increased fitness of wild-born animals, relative to F1 captive-born animals, is that animals caught in the wild have survived early and ongoing natural selection pressures, and are therefore 'fit'. Relaxed selective pressures in captivity mean that even F1 animals that would be 'unfit' in the wild may survive to reproductive age. If these unfit animals are also unproductive, the captive-bred population would exhibit a reduction in reproductive success in the first generation, relative to wild-born animals. In an aquaculture setting this is not necessarily a cause for concern—artificial selection can act to increase population productivity where there is variation in heritable reproductive success over generations of captive breeding. This is most simply demonstrated by the breeder's equation: $R = h^2 S$, where $R$ is the response to selection, $h^2$ is the narrow-sense heritability and $S$ is the selection differential[30,31]. However, as traits linked with broad evolutionary fitness (such as reproductive success) tend to have low heritability[32], selection cannot be relied upon to improve them. Furthermore, processes such as antagonistic pleiotropy can complicate the response to selection on life-history traits in captivity[33]. In the establishment phase of an agricultural program, the potential benefits of long-term selection and domestication must be weighed against short-term productivity losses. In a conservation setting, unintentional selection may be disastrous. For example, offspring survival in utero that differs from Mendelian expectations provides an opportunity for early viability selection that is difficult to prevent and may impact on the effectiveness of pedigree management[34]. Instead, efforts to address variation among breeders during the first generation of captive breeding, such as improved nutrition, should be prioritized[35].

The magnitude of birth-origin effects on reproductive success was influenced by the type of reproductive trait measured. Offspring quality and offspring survival showed the most pronounced decrease of captive-born relative to wild-born reproductive success (Fig. 2). This result indicates the crucial importance of measuring fitness outcomes at multiple life-history stages. Our observation is consistent with a recent meta-analysis[36] that found a close link between offspring quality traits and offspring survival, estimating that a one standard deviation increase of offspring body weight increased survival odds by 71% in mammals and 44% in birds. We observed that fertility/hatchability, reproductive yield and reproductive phenology trait types did not significantly differ between captive-born and wild-born animals, suggesting no evidence that captive-born animals compensate for reduced offspring survival (all lnOR estimates were negative) in a life-history trade-off framework (Table 1). Unnatural social environments or disrupted maternal contact during the early life-stages of captive-born animals may lead to maladaptive development and changes in behavior[6] such as mismothering and offspring abandonment. The mechanisms leading to maladaptive development may explain why we observed a significant decrease in offspring survival without significant differences in other traits that may be less influenced by behavioral changes (e.g., reproductive phenology). Taken together, our results indicate that, if the overall reproductive success of captive

breeding programs is to be improved, population managers would be best placed to focus efforts on improving offspring quality and survival outcomes, as effects on other traits are likely to be weaker. For example, the effect of offspring body weight on juvenile survival in mammals is stronger in captive environments than in wild environments, and offspring mass is positively correlated with maternal mass[36]. Thus, improving maternal nutrition in captive environments may increase offspring quality and survival through increased offspring birth weight.

Reduced offspring survival among captive-bred animals may also result from inbreeding depression—the reduction in fitness as a result of increased homozygosity of inbred animals, and accumulation of deleterious recessive mutations that may be lethal in early life[37]. Captive populations managed for conservation breeding purposes already implement strategies to avoid inbreeding, such as the use of pedigree-based management, and the incorporation of molecular techniques to assist in determining parentage[38]. Likewise, for laboratory research, genomic information is required for long-term management of non-human primate populations[39], and is particularly important not only for preventing inbreeding, but can also reveal genetic variance as a result of mixed ancestry that may influence treatment effects in biomedical research[40]. The avoidance of inbreeding may not be as carefully managed in aquaculture settings, with short-term inbreeding even encouraged to some extent to develop homogenous stock that have uniform body sizes for easier management, and to protect the intellectual property and commercial interests of breeders that supply stock to other fisheries by decreasing genetic variance available for further improvement by selection[41,42]. Aquaculture species are not immune to the effects of inbreeding depression[43], so applying management strategies at a population-level to prevent the effects of inbreeding depression remains a priority for captive breeding programs across all industries.

In this analysis, various measures of reproductive success were included within the offspring survival category, such as juvenile mortality rate, stillbirth/abortion rate and cannibalism, or abandonment of young (Supplementary Table 2). As we have discussed, both genetic (e.g., inbreeding) and non-genetic (e.g., management practices, stress) factors could be responsible for decreased offspring survival in captive-born animals[44–46]. Our dataset precludes determining the cause of this effect; without experimental data it is difficult to disentangle genetic and non-genetic effects. In aquaculture and research environments, we recommend designing experiments to separate these effects, as experimental crosses are more feasible than in conservation programs. For example, Christie et al.[47] identified changes in gene expression between offspring of first-generation hatchery stock of steelhead trout (*Onchorhynchus mykiss*) and offspring of wild stock in captivity, and through a series of crosses were able to rule out maternal effects or chance events. Identifying causative factors will allow captive managers to address these changes, and may inform conservation breeding management. Conservation breeding programs can also benefit from the retrospective analysis of their large detailed datasets in the form of studbooks that are available for many species (see Mason[48] for sources of data). The incorporation of husbandry and behavioral data in regression analyses, possible through the release of the Zoological Information Management System (ZIMS)[49], will assist in determining the factors affecting reproductive success and juvenile mortality.

Our systematic review has identified key areas where the reporting of additional data for captive-breeding studies could be improved, to increase the suitability of these observations for analysis of the effects of captivity in future meta-analysis. In total, 74 comparisons we identified in our systematic review were excluded from the main analysis solely on the criterion of missing

data that precluded calculation of effect sizes. Most commonly, reports of variance (such as the standard deviation or the standard error of the mean), and sample size were missing and unable to be inferred from the text. Together the 12 excluded publications made comparisons involving all four study environments (aquaculture, research, conservation and other), and all five trait type categories. The excluded data covered 10 additional species not otherwise included in our meta-analysis (Supplementary Table 5), which were not taxonomically distinct from other species included in our analysis. Our results did not change greatly with the inclusion of 17 of these comparisons using multiple imputation (Supplementary Table 4). The call for careful reporting of all relevant statistics required for meta-analysis in primary studies has been made often; recently Gerstner et al.[50] provided a useful guide to authors as to what to include.

In conclusion, our meta-analysis shows that wild-born animals generally have higher reproductive success than their captive-born counterparts in captive environments, across multiple industries and irrespective of taxonomy. The increased reproductive success of wild-born relative to captive-born animals was particularly evident in aquaculture environments, which were more likely to report wild versus first-generation comparisons than studies from other environments. We urge greater reporting of the general characteristics of captive population studies, in particular generations of captive breeding, to enable a greater understanding of effects at the first and subsequent generations. Our literature search uncovered a large body of literature on other types of captive to wild comparisons that were not the target of our search criteria (Supplementary Fig. 2) and which therefore cannot be considered a systematic survey. Nevertheless, future systematic searches into these areas, especially captive-born to wild-born animals in the wild (e.g., reintroductions) may reveal long-term effects of captive breeding. Now that we have found strong evidence of birth-origin effects on reproductive success within captive environments, future research should experimentally investigate the factors driving these changes, to inform management decisions, such as preventing adaptation to captivity, avoiding inbreeding, reducing juvenile mortality, and establishing successful closed-cycle breeding programs.

## Methods

**Data collection**. Following the PRISMA guidelines for systematic reviews and meta-analyses[51,52], we searched the 'Web of Science' database on 26 April 2016 and the 'Scopus' database on 7 June 2016, with no language or time restrictions, using the following terms related to reproductive traits and birth origin: (reproduct* OR product* OR hatch* OR fecund* OR "breeding success" OR "litter size" OR "juvenile mortality" OR "infant mortality") AND (captiv* OR "zoo-born") AND ("wild-born" OR "wild-caught" OR "wild-laid" OR "wild-bred" OR "free-ranging"). We also screened reference lists in relevant papers to obtain the broadest possible coverage. We obtained 1065 results from our search of the 'Web of Science', and 600 results from our 'Scopus' search.

See Supplementary Fig. 2 for the overview and outcomes of our search strategy. We first removed duplicates between and within the two databases, leaving 1160 unique works. We next examined the abstract and title of all works to identify potentially relevant primary sources, and downloaded full texts of sources that appeared to meet our inclusion criteria (see below). We considered only published papers that were the primary source of data (i.e., excluded reviews, books, conference proceedings and syntheses) to avoid the duplication of reporting. In order to isolate papers on our research topic of interest (the effects of birth origin in captivity), we classified all papers by the study populations they compared:

a.  wild-born vs. captive-born in captivity (comparison of interest),
b.  wild-born vs. captive-born in the wild (such as in reintroductions),
c.  wild-born in captivity compared to the wild,
d.  wild populations compared to captive populations
e.  other comparisons (such as those made at the level of the ancestors), and
f.  two or more of the above.

Of 1160 papers examined, 126 (10.9% of unique results) were screened by two people to ensure agreement and minimize the risk of researcher bias. After grouping papers by study population (i.e., a—f, above), papers that compared wild-born to captive-born animals in captivity (category a, and 12 papers from category

f, total $N = 125$) were further screened to identify studies that reported data for at least one reproductive trait ('comparison', used for calculating the effect sizes used in our meta-analysis). A total of 56 papers comparing reproductive traits in wild-born and captive-born animals in captivity were identified, encompassing 242 wild-born/captive-born comparisons considered for inclusion in the study.

**Data extraction**. We developed a data coding strategy to classify the comparisons by recording the first author, year of publication, journal of publication, species of study (common name and scientific name), study environment (see below), whether the captive generation of comparison was specified (e.g., F1, F2), comparison (reproductive trait) for each study, trait type (see below), measurement/statistic, error, and sample size.

Study environment was determined from the reported purpose for keeping the captive population, as described by the authors of each publication, and categorized as either:

a.  Aquaculture—may occur in the laboratory, but the primary purpose is for commercial production/domestication of animals for consumption or trade
b.  Conservation—a captive breeding program with the purpose of propagating the species to reinforce the wild population, to provide an insurance population against extinction in the wild, or to educate members of the public
c.  Research—the purpose of the captive program is to provide animals for research under controlled conditions, for reasons other than developing a closed life-cycle production system, unless this is for a laboratory research species. The results of the study may inform conservation outcomes, but the animals are not propagated for conservation purposes
d.  Other—does not fall into any of the above categories

Studies were included in the review and meta-analysis if they fulfilled the following inclusion criteria: (i) studies must have made at least one comparison of a reproductive trait between captive-born and wild-born animals of any species in a captive environment. Some studies hold animals in 'semi-natural' enclosures—we considered a study to take place in a captive environment if there were human barriers to movement for the purpose of holding animals and if some form of provisioning of resources (such as shelter, food and/or water) occurred. We did not require that the animals in a study were housed in the same physical location to be included in the meta-analysis, as long as the enclosure types were similar. For example captive-born and wild-born animals of the same species across multiple zoos were included. We considered animals to be 'wild-born' if they were brought into captivity from the wild either as eggs, young or mature individuals. (ii) Studies did not duplicate other included studies. In cases where duplicates were identified (by species studied, population reported, years of analysis and sample size), we selected the study that was most recent, or which had the greatest sample size ($N = 1$ study comprising 1 comparison was excluded for this reason). (iii) Papers (including any supplementary material) must contain extractable data (for example means, standard deviations and sample sizes, or other statistics or raw data that could be used to calculate effect sizes), this criterion resulted in the exclusion of $N = 74$ comparisons. (iv) The study must not have experimentally manipulated reproductive success, for example through the restriction of diet ($N = 4$ comparisons excluded). (v) Reproductive success was not systematically influenced by bias in opportunity to breed ($N = 22$ comparisons excluded). For example, many comparisons such as lifetime reproductive output can be influenced by captive management if wild-born animals are prioritized for breeding over captive-born animals as is the case in conservation breeding programs that aim to maximize the genetic contribution of founder animals[53]. (vi) Data were not duplicated within the study ($N = 5$ comparisons). For example, if male reproductive success, female reproductive success, and overall reproductive success were reported, only overall reproductive success was included. The excluded studies and the reasons for their exclusion are given in Supplementary Table 5, with a flowchart of data filtering provided in Supplementary Fig. 3.

As our studies included diverse species and breeding strategies, we obtained many different comparisons related to reproductive success or failure. These were broadly categorized into 'trait types' as comparisons relating to the following: fertility/hatchability, reproductive yield, offspring quality, offspring survival, and reproductive phenology (Supplementary Table 2). For each comparison, we determined whether it had a positive or negative relationship with overall reproductive success (Supplementary Table 2). An increase in a comparison with a positive relationship would result in increased reproductive success. For example, an increase in probability of breeding, fertility rate, hatching rate, juvenile survival rate, and litter/clutch/spawn size are expected to be typically positively correlated with reproductive success. Increased interbirth intervals, juvenile mortality and age at first parturition are expected to be typically negatively correlated with productivity. For other comparisons, the directionality of a relationship with reproductive success was unclear, for example date of parturition, gestation length and offspring sex ratio. As such, comparisons for which the direction of the effect on overall reproductive success could not be characterized were excluded from the meta-analysis ($N = 20$ comparisons).

After filtering on our inclusion criteria, 39 papers contributed 115 reproductive comparisons between captive-born and wild-born animals in captive environments for analysis. All 39 papers were coded by the same person, with 18 of these (46%) coded by an additional person to ensure agreement with the coding strategy.

**Effect size extraction and calculation**. For each comparison that satisfied our inclusion criteria, we extracted raw data for both the captive-born and the wild-born population reported in the text or in tables/figures (including Supplementary Material) to calculate an effect size, a measure of the magnitude and direction of the difference between the two populations (detailed below). Data that were reported only graphically were extracted using GetData Graph Digitizer 2.26[54]. For continuous comparisons (such as number of offspring), we obtained the mean, standard deviation and sample size for each group. Where the standard error was the only variance measure reported, we calculated the standard deviation as $SD = SE \times \sqrt{N}$. If only 95% confidence intervals were presented, we calculated the standard deviation as $SD = \sqrt{N} \times \frac{(\text{upper } 95\% \text{ CI} - \text{lower } 95\% \text{ CI})}{3.92}$. For proportional comparisons (such as hatching rate), we recorded the number of events out of the total sample size ($n/N$). Some studies reported the frequency of singletons, twins, triplets, and quadruplets between wild-born and captive-born animals in captivity. Where possible, overall litter size was calculated from this data and used as the comparison instead.

Stochastic dependency can occur when multiple comparisons are made of the same data, resulting in biased wild-born to captive-born comparisons[55,56]. In our dataset, this non-independence occurs in studies that reported productivity for the population of wild-born individuals, compared multiply to each generation of captive breeding. In such cases, we obtained the overall mean for the captive-born animals for effect size calculation, where possible. If overall values were not calculable, we used only the data from the first generation of captive breeding (F1) to compare to the wild-born generation (F0). Likewise, for studies that compared more than two populations (e.g., wild-born animals compared to two groups of captive-born animals), we included the effect size associated with only one comparison, chosen as the pair of populations most comparable to one-another in all other respects (e.g., housed at the same location under the same conditions, comparison reported for the same year), or by pooling data from the multiple captive-born populations if they were identical treatments (e.g., tanks of fish).

We chose the log odds ratio (lnOR) as our measure of effect size, as it could be calculated for both the continuous and proportional data present in our analysis. Log odds ratios between the wild-born population and captive-born population and their unbiased estimates of sampling variances were computed for each comparison using the 'metafor' package in R[57,58]. The log odds ratio is a symmetric measure centered around zero; data were input such that a positive log odds ratio refers to increased reproductive success of captive-born animals relative to wild-born counterparts and a negative log odds ratio refers to the converse. A small constant (0.5) was added to zero values in proportional data to allow for estimation of the effect size; this applied to 3/48 (6%) of effect sizes calculated from proportional data.

**Meta-analytic procedures**. To account for the non-independence of effect sizes as a result of the shared evolutionary history of closely related species, we obtained the phylogenetic correlation between the species in our meta-analysis using the 'rotl' package[59] in R, based on published phylogenies available through the Open Tree of Life[60]. Taxon names were matched to records in the Open Tree Taxonomy, to obtain relationships between species. *Chironex fleckeri* was used as the outgroup to obtain the full variance-covariance matrix of phylogenetic relationships. Due to the diverse species in our meta-analysis, accurately estimating branch lengths was not plausible, so we computed branch lengths based on topology (Fig. 1) using the 'ape' package[61] in R.

We fitted multi-level hierarchical models in the 'MCMCglmm' package[62] in R. Each model was run for $5 \times 10^6$ iterations, with a burn-in of $1.5 \times 10^5$ and a thinning interval of 3000, with an inverse-gamma prior ($V = 1$, $nu = 0.002$). We report the posterior mode and the 95% highest posterior density credible intervals (95% HPD CIs) for each model set. Model diagnostics were checked so that autocorrelation <0.1. Chain convergence was confirmed visually by passing the Heidelberg stationarity test and by a Gelman-Rubin statistic <1.1 based on three runs of each model.

We performed sensitivity analyses by comparing the overall model (with study ID as a random effect) to one with both study ID and phylogeny as random effects. We considered the model with the lowest Deviance Information Criterion (DIC) value the best model. Cohen's established recommendations for the interpretation of small (Pearson's correlation coefficient $\phi = 0.1$), medium ($\phi = 0.3$) and large ($\phi = 0.5$) effects are equivalent to odds ratios of 1.22, 1.86, and 3.00 with equal treatment-control sample sizes[63,64]. These correspond to estimates of log odds ratios from our models of ±0.20, 0.62, and 1.10 as small, medium, and large, respectively. Estimates with a 95% HPD CI excluding zero were taken as statistically significant at $\alpha = 0.05$. We note that these benchmarks do not establish biological importance[65], so we discuss our results in terms of their practical implications for captive breeding programs.

Traditional calculations of heterogeneity such as $I^2$ assume that effect sizes are independent, however this is not the case for multi-level models so the extended heterogeneity statistic was instead calculated following Nakagawa & Santos[66]. Doing so enabled us to partition total heterogeneity ($I^2_{total}$) into phylogenetic variance ($I^2_{phylogeny}$), study ID variance ($I^2_{study}$) and residual variance ($I^2_{residual}$). Heterogeneity well above the $I^2_{total} > 75\%$ benchmark for high heterogeneity[67] is common across ecological and evolutionary meta-analyses[26]. For the phylogenetic model, we obtained lambda, a measure of phylogenetic signal or phylogenetic heritability ($H^2$), where $H^2 = 0$ indicates no phylogenetic relatedness among effect sizes[68]. As both our models had high heterogeneity, but phylogenetic signal was low, we proceeded with non-phylogenetic meta-regression models to fit moderators including 'captive environment' and 'trait type'.

**Publication bias**. We assessed publication bias in our meta-analysis using three methods. First, we fitted a non-phylogenetic meta-regression model with year of publication as a moderator. Evidence of time-lag bias is indicated if the 95% HPD CI of the slope estimate excludes zero. Second, we used funnel plots to visualize possible publication bias (evident by funnel plot asymmetry), by plotting the effect sizes and the meta-analytic residuals against their precision ($\sqrt{\frac{1}{variance}}$). Funnel plot asymmetry can also result from high heterogeneity, so applying these publication bias tests to the meta-analytic residuals instead of the raw effect sizes minimizes the effect of heterogeneity on funnel plot asymmetry[66]. Third, we performed Egger's regression[69] on the meta-analytic residuals obtained from the overall model to formally test for evidence of funnel plot asymmetry, by fitting a linear model of the meta-analytic residuals against their precision. If the intercept of the Egger's regression is significantly different from 0 (at $\alpha = 0.05$), publication bias may be present. We then performed a trim-and-fill analysis using the 'trimfill' function in 'metafor' to estimate the number of effect sizes potentially missing from our dataset. Finally, all models were rerun on a subset of the dataset after outliers identified in the funnel plot were removed, to examine whether any model results changed substantially. Publication bias results are presented in Supplementary Note 1 and Supplementary Fig. 1a–c.

**Multiple imputation**. We performed multiple imputation to recover missing data. Of the 74 comparisons excluded for missing data, 17 of these were continuous traits with the mean reported but not the standard deviation, and which could be estimated. We performed 20 imputations using the 'mice' function in the R package 'mice'[70] and re-ran all models with each imputation. The posterior mode of the pooled posterior distributions from each imputed meta-analysis was used for inference and qualitatively compared to the main dataset results to check support for our conclusions. We also considered the effect of imputing missing sample sizes for a further 24 comparisons. Full multiple imputation details and results are presented in Supplementary Note 2 and Supplementary Fig. 4. R-code for all analyses reported herein is included in Supplementary Data 2.

**Data availability**. All data associated with this manuscript has been uploaded as Supplementary Information (Supplementary Data 1); contact the corresponding author for further information.

**Code availability**. Code associated with the analyses performed in this manuscript has been uploaded as Supplementary Information (Supplementary Data 2); contact the corresponding author for further information.

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

## Acknowledgements

This research was supported by the Australian Research Council Discovery Projects grant DP170101253 to C.E.G, San Diego Zoo Global, The University of Sydney and by an Australian Government Research Training Program (RTP) Scholarship to KAF. We are grateful to Angie Jarman (Angie's Animal Art) for providing the line drawings used in Fig. 1. We also thank Barbara Streibl (University of Tübingen) for assistance with literature filtering.

## Author contributions

K.A.F. collected the data, analyzed the data, prepared the figures, and drafted the manuscript. C.J.H. provided conceptual guidance on the analysis and critically revised

the manuscript. C.E.G. conceived the project, provided technical guidance on the analysis, and critically revised the manuscript.

## Additional information

**Competing interests:** The authors declare no competing interests.

