## [Peer Review File · Nature Communications]

Reviewers' comments:

Reviewer #1 (Remarks to the Author):

A meta-analysis of productivity differences in phylogenetically diverse captive environments

Farquharson et al.

I have read this meta-analysis with great interest. This meta-analysis looks at the effects of captive breeding on aspects of offspring production and fitness, which have profound implications in food production, conservation and also animal ethics. They found that overall, wild-born animals do significantly better than captive counterparts. Interestingly, further analysis revealed that offspring survival and quality are severely compromised in captive-born animals.

I have read many meta-analyses before and I must say this meta-analysis has been conducted at the highest level. And the paper is very well written. However, my one major concern is that the authors excluded a lot of studies due to some missing information. They say

"In total, 72 comparisons we identified in our systematic review were excluded solely on the criterion of missing data that precluded calculation of effect sizes. Most commonly, measures of variance (such as the standard deviation or the standard error of the mean), and the sample size were missing and unable to be inferred. The 12 publications excluded due at least in part to missing data covered 10 additional species not otherwise included in our meta-analysis (Supplementary Table 4)."

The missing information is mainly due to missing variance (SE, SD) or sample sizes. Actually, you can recover these types of information fairly reliably, using missing data imputation (multiple imputation, MI). Given including these studies with missing information dramatically increase this meta-analysis's sample size, I feel it is important to do MI at least as a part of sensitivity analysis. This meta-analysis uses a MI approach.

Besson, A. A., Lagisz, M., Senior A. M., Hector, K. L. & Nakagawa, S. (2016) Effect of maternal diet on offspring coping styles in rodents: a systematic review and meta-analysis. *Biological Reviews*. 91: 1065-1080

The code is available here

https://github.com/mlagisz/MA_maternal_diet_coping_styles

Also, these chapters are useful for implementing MI and how you may do so in the context of meta-analysis.

Nakagawa, S. (2015) Missing data: mechanisms, methods and messages In: *Ecological Statistics: contemporary theory and application* (eds. Fox, G. A., Negrete-Yankelevich, S. & Sosa, V. J.). Oxford University Press, Oxford. pp. 81-105

Gurevitch J. & Nakagawa, S. (2015) Research synthesis methods in ecology In: *Ecological Statistics: contemporary theory and application* (eds. Fox, G. A., Negrete-Yankelevich, S. & Sosa, V. J.). Oxford University Press, Oxford. pp. 201-228

As written above, I think by using MI, the authors can recover many of these missing information and thus, they will be able to increase the power of their meta-analysis significantly.

Reviewer #2 (Remarks to the Author):

This is an important paper that covers an issue crucial to many conservation/ production sectors, and that potentially has animal welfare implications too. The topic is also very scientifically interesting. The analysis was very ambitious in scope: comprehensive and bold. It also seems technically very sophisticated, with the use of PCMs especially to be welcomed (although we have to admit some of the statistics were beyond our expertise, especially the use of logistic regressions instead of more conventional models arguably more clearly designed for use with continuous variables). In terms of presentation, the paper was clearly written and through: it was logically laid out, and well referenced, with the introduction being particularly informative (giving a good background on the three industries focused on and the types of goals and problems these currently face). We had just three major comments, and then a number of small ones that should be very easy to address.

Major comments

The results of the paper may not support the broad conclusion that captive-born (CB) animals were "less productive across various captive environments" (e.g. as stated in the abstract). When the authors split the environments, only animals in the commercial sectors were significantly affected by birth origin. This suggests that the effect in commercial environments has considerable leverage, 'pulling' the combined group "captive environments" in that direction when the three environments are pooled together. We therefore feel it would be more fairly representative of the results to emphasise the strong significance of the effects in commercial category and then separately mention the weaker negative trends in the conservation and research groups.

Commercial usage was confounded with no. of generations (being skewed towards F1), as the authors highlight themselves but apparently do not tease out analytically. Furthermore, environment was also confounded with taxa, the commercial group being the only one to contain invertebrates and also the only one to contain no homeotherms. This is not acknowledged in the paper. We suggest that the 'commercial' group is relabeled 'aquaculture', and the confound between animal type and environment type thus at least implicitly acknowledged. We would further urge that CB generation no. is factored in statistically, to try and partial out its effects.

Finally, some issues require clarification and /or justification throughout the text. First, 'productivity' was not clearly defined in the text itself (though more details are provided in the SOM). In abstract it is defined as overall reproductive success, but this is quite vague, and as far as we can tell 'lifetime reproductive success' or 'number of grandchildren' are never the metrics used: all measures concerned the rate of output of viable young (high rates yielding high scores even if an animal's reproductive period is short-lived and/or its infants poor at reproducing themselves). Fig. 2 lists the "trait types" examined, but a fuller explanation of what "productivity" encompasses would still be welcome in the text. What exactly IS "breeding success" for example (especially as to some audiences, breeding means mating, not producing progeny). Likewise, for juvenile survival rate: what did this cover (anything between birth/hatching and maturity? Or the authors' own definitions in their papers?). A second aspect we felt could be better defined was what it took to be considered wild-born: were animals still considered wild-born if they were brought into captivity when very young and most of their development took place there? What if they developed from eggs fertilised in and harvested from the wild? Overall, was there a cut off for being judged as being "wild born"? Lastly, when examining wild-born and captive-born animals in shared captive environments (as mentioned in e.g line 42), in order to be included does a study have to examine both populations in the same location/captive environment, or are studies included that compare populations from different

locations? Clarification was needed on this point too.

Minor comments

- Might well want to mention birth origin (or something related) in the title; and it's surely the animals that are "phylogenetically diverse", not the environments?

- Could the other potential comparisons yielded by the original literature search (e.g. captive bred in the wild versus captivity) also have yielded some insights into potential mechanism, and/or 'benchmarking' values for revealing whether birth origin differences reflect abnormally high productivity in wild-born captive animals versus instead abnormally low in captive-born captive animals?

- Should animals that hatch from eggs even be termed "born"?

- Lines 30-37: when discussing the need for closed cycle breeding programs the authors could touch on how adjusting to a captive environment is very stressful for wild-born individuals from a welfare perspective, which adds to the importance of these programs.

- Well thought out discussion, but it might be useful to add a direction for future research that suggests examining productivity of captive born animals in the wild. Breeding for animals that have high productivity in captivity may not actually be helpful if the end goal is reintroduction.

- Line 229-238: possibly include an example or reference that might offer an explanation to why wild-born populations have higher offspring quality and survival to help bridge the gap between the increase in these two traits while there is no significant difference in breeding success, reproductive output and rate of reproduction. Or even a hypothesis or theory on the phenomenon. Note that Mason et al. '13 discuss a range of possible mechanisms especially for changes manifest in F1, if useful (Mason, G., Burn, C. C., Dallaire, J. A., Kroshko, J., Kinkaid, H. M., & Jeschke, J. M. (2013). Plastic animals in cages: behavioural flexibility and responses to captivity. *Animal Behaviour*, 85(5), 1113-1126.)

- Why take the time to organize all of the papers into 6 categories of populations compared if only looking at wild-born vs. captive-born animals in captivity?

- Line 407-408: "Of these papers, 18 (46%) were coded by 2 people for consistency." → not sure if we're missing something, but why only 18 of the papers? Did the rest not need to be coded, or just not by 2 people?

- SOM: Exclusion of Clubb et al.'s elephant work for poor reporting: in fact this work did not even attempt to look at birth origin effects on reproduction, because sample sizes in the CB group were too small (very few CB elephants in zoos had reproduced at the time of the study).

As a final note, we should add that we very much look forward to seeing this great work published!

Georgia Mason and three graduate students (Aileen McLennan, Sam Decker and Miranda Bandeli)

Reviewer #3 (Remarks to the Author):

Review of "A meta-analysis of productivity differences in phylogenetically diverse captive environments." In this manuscript, the authors find that, across all captive breeding programs wild-origin broodstock have higher productivity in captive environments than do hatchery-origin broodstock. A number of comments follow that will hopefully be helpful in revising the manuscript.

Comments:

1. It is unclear what is gained by combining analyses across so many different types of captive breeding programs that have entirely different aims. The goals associated with captive breeding programs for producing organisms for research (e.g., zebrafish) are entirely different than the goals of captive breeding programs that focus on conservation. If productivity is lower in a research program, but not a conservation program, is that meaningful? Furthermore, the definition of "productivity" is a little unclear. Sometimes it means reproductive success and other times it means traits are (assumed to be) correlated with reproductive success.

2. As the authors point out, it cannot be determined if the reductions in productivity (e.g., reproductive success) are due to genetic effects or environmental effects. If the effects are entirely environmental, then the implications of these findings are perhaps overstated because a simple change in the captive program (e.g., diet) could greatly increase productivity.

3. It is clear that there is a decrease in productivity for captive-born broodstock when looking at the log odds ratio, but it is unclear if it is biologically meaningful. How do these results translate to reproductive success? If captive-born broodstock produce 0.5% fewer offspring than wild-born offspring is that meaningful?

Minor comments:

Line 20-22: It is unclear how these results translate into a larger issue. What do these findings really mean?

Line 24: And elsewhere, the definition of productivity is not always consistent.

Line 62: Maternal effects can have a genetic basis.

Lines 63-66: I could not follow what is trying to be conveyed in this sentence.

Line 77: All captive breeding programs do not aim to maximize productivity!

Line 90: It is unclear to me what purpose Figure 1 serves.

Line 106: It would be useful to explain up front why the number of comparisons is >> than the number of studies.

Line 148: Why do you believe this?

Could results be inflated by single outlier studies?

Line 205: Maybe. It depends on the environment and the context - there is a large body of literature on maladaptation.

Lines 210-225: Much of this text could be omitted or re-written to be clearer.

Lines 317-321: I think you may have missed a number of studies using only those terms.

Line 346: What do you mean by "data coding strategy"?

Author response to reviewer comments, NCOMMS-17-19589
"A meta-analysis of birth-origin effects on reproduction in diverse captive environments"

Reviewer #1 (Remarks to the Author):

A meta-analysis of productivity differences in phylogenetically diverse captive environments
Farquharson et al.

I have read this meta-analysis with great interest. This meta-analysis looks at the effects of captive breeding on aspects of offspring production and fitness, which have profound implications in food production, conservation and also animal ethics. They found that overall, wild-born animals do significantly better than captive counterparts. Interestingly, further analysis revealed that offspring survival and quality are severely comprised in captive-born animals.

I have read many meta-analyses before and I must say this meta-analysis has been conducted at the highest level. And the paper is very well written. However, my one major concern is that the authors excluded a lot of studies due to some missing information. They say

"In total, 72 comparisons we identified in our systematic review were excluded solely on the criterion of missing data that precluded calculation of effect sizes. Most commonly, measures of variance (such as the standard deviation or the standard error of the mean), and the sample size were missing and unable to be inferred. The 12 publications excluded due at least in part to missing data covered 10 additional species not otherwise included in our meta-analysis (Supplementary Table 4)."

The missing information is mainly due to missing variance (SE, SD) or sample sizes. Actually, you can recover these types of information fairly reliably, using missing data imputation (multiple imputation, MI). Given including these studies with missing information dramatically increase this meta-analysis's sample size, I feel it is important to do MI at least as a part of sensitivity analysis. This meta-analysis uses a MI approach.

Besson, A. A., Lagisz, M., Senior A. M., Hector, K. L. & Nakagawa, S. (2016) Effect of maternal diet on offspring coping styles in rodents: a systematic review and meta-analysis. *Biological Reviews*. 91: 1065-1080

The code is available here
https://github.com/mlagisz/MA_maternal_diet_coping_styles

Also, these chapters are useful for implementing MI and how you may do so in the context of meta-analysis.

Nakagawa, S. (2015) Missing data: mechanisms, methods and messages In: *Ecological Statistics: contemporary theory and application* (eds. Fox, G. A., Negrete-Yankelevich, S. & Sosa, V. J.). Oxford University Press, Oxford. pp. 81-105

Gurevitch J. & Nakagawa, S. (2015) Research synthesis methods in ecology In: *Ecological Statistics: contemporary theory and application* (eds. Fox, G. A., Negrete-Yankelevich, S. & Sosa, V. J.). Oxford University Press, Oxford. pp. 201-228

As written above, I think by using MI, the authors can recover many of these missing information and thus, they will be able to increase the power of their meta-analysis significantly.

Response: Thank you for this helpful suggestion. Of the 74 comparisons excluded for missing data, 17 with missing standard deviations were able to be recovered through multiple imputation. We performed a sensitivity analysis to compare the results from our original models (N = 115 comparisons) to results from models with MI data included (N = 132 comparisons). The results of both sets of analyses were similar – the overall model, aquaculture environment and offspring quality and offspring survival traits remained statistically significant and the estimated effects were of similar magnitude across all analyses in each of the two datasets. The MI dataset did not greatly increase the power of our meta-analysis, as only 17 comparisons could be recovered and imputation increased uncertainty around our estimated effects (Supplementary Table 5). To increase power, we also imputed missing sample sizes of an additional 24 comparisons (N = 156 in

total). The estimated effects were in the same direction and of similar magnitude to the original analysis, however we are not confident in the relationship between the mean and sample size used to impute this data (Supplementary Fig. 4). As such, we report the results of the analysis with missing standard deviations imputed (N = 132) in the Supplementary, as there was a clear relationship between the mean and standard deviation (Supplementary Fig. 4).

Despite the increased uncertainty, because the estimated effects were similar across our two datasets, we do not believe that our overall conclusions are biased by missing data. We now report the methods, results and discussion of our MI analysis in full in the Supplementary Text, and in Supplementary Table 5. We have also reported a brief summary of our methods (L541-548) and results (L167-173) in the main text.

Reviewer #2 (Remarks to the Author):

This is an important paper that covers an issue crucial to many conservation/ production sectors, and that potentially has animal welfare implications too. The topic is also very scientifically interesting. The analysis was very ambitious in scope: comprehensive and bold. It also seems technically very sophisticated, with the use of PCMs especially to be welcomed (although we have to admit some of the statistics were beyond our expertise, especially the use of logistic regressions instead of more conventional models arguably more clearly designed for use with continuous variables). In terms of presentation, the paper was clearly written and through: it was logically laid out, and well referenced, with the introduction being particularly informative (giving a good background on the three industries focused on and the types of goals and problems these currently face). We had just three major comments, and then a number of small ones that should be very easy to address.

Major comments

The results of the paper may not support the broad conclusion that captive-born (CB) animals were "less productive across various captive environments" (e.g. as stated in the abstract). When the authors split the environments, only animals in the commercial sectors were significantly affected by birth origin. This suggests that the effect in commercial environments has considerable leverage, 'pulling' the combined group "captive environments" in that direction when the three environments are pooled together. We therefore feel it would be more fairly representative of the results to emphasise the strong significance of the effects in commercial category and then separately mention the weaker negative trends in the conservation and research groups.

Response: The overall result of our analysis across all environments is significant, and all effects across the aquaculture, conservation and research environments were in the same negative direction, though aquaculture was the only strong significant effect. We have now emphasized the differences in the abstract *"Our meta-analysis examining 115 effect sizes from 44 species of invertebrates, fish, birds, and mammals showed that, overall, captive-born animals have a 42% decreased odds of reproductive success in captivity compared to their wild-born counterparts. The largest effects were seen in commercial aquaculture, relative to conservation and laboratory settings, and offspring survival and offspring quality were the most sensitive traits in our dataset. Although a somewhat weaker trend, reproductive success in conservation and laboratory research breeding programs was also in a negative direction for captive-born animals."* (L16-20).

Commercial usage was confounded with no. of generations (being skewed towards F1), as the authors highlight themselves but apparently do not tease out analytically. Furthermore, environment was also confounded with taxa, the commercial group being the only one to contain invertebrates and also the only one to contain no homeotherms. This is not acknowledged in the paper. We suggest that the 'commercial' group is relabeled 'aquaculture', and the confound between animal type and environment type thus at least implicitly acknowledged. We would further urge that CB generation no. is factored in statistically, to try and partial out its effects.

Response: We have renamed the "commercial production" category to "aquaculture" to acknowledge the confound with species and environment. Controlling for phylogeny in our analysis did not substantially change our overall results (reported at L106), so although the aquaculture environment was more phylogenetically distinct from the other groups (there was some overlap, as some fish species were included in the "conservation" category), we do not believe that the taxonomy is driving our results.

We agree with the reviewer that the relationship between number of generations and captive environment is an important area for investigation, and ideally would be statistically analyzed in a meta-regression model. Disappointingly, the vast majority of studies we examined did not report the generation number of the captive-born animals at all (80 out of 115 comparisons did not specify the captive-born generation), as we indicated in Supplementary Table 3 and in the main text (L158-164). We were therefore not able to statistically analyze whether differences in reproductive success in the aquaculture environment are as a result of the characteristics of the captive environment or of first generation changes, which we have now clarified (L160). We recommend in the Discussion (L328) that future studies in captive environments report the number of generations in captivity so that these important effects can be disentangled.

Finally, some issues require clarification and /or justification throughout the text. First, 'productivity' was not clearly defined in the text itself (though more details are provided in the SOM). In abstract it is defined as overall reproductive success, but this is quite vague, and as far as we can tell 'lifetime reproductive success' or 'number of grandchildren' are never the metrics used: all measures concerned the rate of output of viable young (high rates yielding high scores even if an animal's reproductive period is short-lived and/or its infants poor at reproducing themselves). Fig. 2 lists the "trait types" examined, but a fuller explanation of what "productivity" encompasses would still be welcome in the text. What exactly IS "breeding success" for example (especially as to some audiences, breeding means mating, not producing progeny). Likewise, for juvenile survival rate: what did this cover (anything between birth/hatching and maturity? Or the authors' own definitions in their papers?). A second aspect we felt could be better defined was what it took to be considered wild-born: were animals still considered wild-born if they were brought into captivity when very young and most of their development took place there? What if they developed from eggs fertilised in and harvested from the wild? Overall, was there a cut off for being judged as being "wild born"? Lastly, when examining wild-born and captive-born animals in shared captive environments (as mentioned in e.g line 42), in order to be included does a study have to examine both populations in the same location/captive environment, or are studies included that compare populations from different locations? Clarification was needed on this point too.

Response: We accept that our terminology was ambiguous, and have thought carefully about what terms to use that accurately describe the diverse data types that are included in our analysis. To that end, we have removed the term "productivity" in the text and instead used "reproductive success" throughout, which we define at L81: "*We take a broad definition of 'reproductive success' to refer to diverse measures of reproductive traits, encompassing production of gametes/offspring at multiple stages throughout the life history of breeders*". We have also renamed some of our reproductive trait categories, to be more specific. Specifically, the "breeding success" category is now "fertility/hatchability", the "reproductive output" category is "reproductive yield" and the "rate of reproduction" category is now "reproductive phenology". Our categories are broad to reflect the diverse measurements of reproductive success in the papers we examined. We note that all traits are provided at Supplementary Table 2, and we now clarify in the table caption that these traits are recorded as defined by the original authors each study.

In addition, we have clarified our definition of "wild-born" in the methods: "*We considered animals to be 'wild-born' if they were brought into captivity from the wild either as eggs, young or mature individuals.*" (L402).

We have removed reference to a shared captive environment (L44). We originally referred to shared captive environments in a general sense to highlight that differences may occur as a result of birth origin rather than different environments. We have clarified our selection criteria in the Methods, "*We did not require that the animals in a study were housed in the same physical location to be included in the meta-analysis, as long as the enclosure types were similar. For example, captive-born and wild-born animals of the same species across multiple zoos were included.*" (L399-401).

Minor comments

- Might well want to mention birth origin (or something related) in the title; and it's surely the animals that are "phylogenetically diverse", not the environments?

Response: Good point; we have changed title to "*A meta-analysis of birth-origin effects on reproduction in diverse captive environments*".

- Could the other potential comparisons yielded by the original literature search (e.g. captive bred in the wild versus captivity) also have yielded some insights into potential mechanism, and/or 'benchmarking' values for revealing whether birth origin differences reflect abnormally high productivity in wild-born captive animals versus instead abnormally low in captive-born captive animals?

Response: Our search criteria were targeted towards comparisons made in captive environments, so this was the focus of our systematic review. We reported information about the other types of comparisons to give a background to other research in this area, which was opportunistically captured by our search criteria. We expect that investigating differences in these other environments may also be of interest, particularly with respect to conservation reintroductions, however this would require a separate systematic review of the literature, with appropriately targeted search terms. We now comment on these opportunities in the Discussion (L329-334), *"Our literature search uncovered a large body of literature on other types of captive to wild comparisons that were not the target of our search criteria (Supplementary Figure 2) and which therefore cannot be considered a systematic survey. Nevertheless, future systematic searches into these areas, especially captive-born to wild-born animals in the wild (e.g. reintroductions) may reveal long-term effects of captive breeding"*.

- Should animals that hatch from eggs even be termed "born"?

Response: Our systematic review includes diverse species with different breeding strategies, including birds, invertebrates, fish, marsupials and eutherian mammals. As indicated above, we have now defined the term "wild-born" (L402).

- Lines 30-37: when discussing the need for closed cycle breeding programs the authors could touch on how adjusting to a captive environment is very stressful for wild-born individuals from a welfare perspective, which adds to the importance of these programs.

Response: This is a good point, and we have now added this welfare perspective: *"Successful captive breeding, as opposed to continual supplementation of captive populations with wild animals, can also help avoid additional welfare concerns arising from wild-born animals adjusting to a captive environment (Mason et al. 2013)"* (L36).

- Well thought out discussion, but it might be useful to add a direction for future research that suggests examining productivity of captive born animals in the wild. Breeding for animals that have high productivity in captivity may not actually be helpful if the end goal is reintroduction.

Response: As indicated above, we have added this potential future direction (L329-334).

- Line 229-238: possibly include an example or reference that might offer an explanation to why wild-born populations have higher offspring quality and survival to help bridge the gap between the increase in these two traits while there is no significant difference in breeding success, reproductive output and rate of reproduction. Or even a hypothesis or theory on the phenomenon. Note that Mason et al. '13 discuss a range of possible mechanisms especially for changes manifest in F1, if useful (Mason, G., Burn, C. C., Dallaire, J. A., Kroshko, J., Kinkaid, H. M., & Jeschke, J. M. (2013). Plastic animals in cages: behavioural flexibility and responses to captivity. *Animal Behaviour*, 85(5), 1113-1126.)

Response: This is an interesting phenomenon, and we thank the reviewer for pointing out this paper. We have added this citation, along with the statement that *"Unnatural social environments or disrupted maternal contact during the early life-stages of captive-born animals may lead to maladaptive development and changes in behavior (Mason et al. 2013) such as mismothering and offspring abandonment. The mechanisms leading to maladaptive development may explain why we observed a significant decrease in offspring survival without significant differences in other traits that may be less influenced by behavioral changes (e.g. reproductive phenology traits)"* (L256-261).

- Why take the time to organize all of the papers into 6 categories of populations compared if only looking at wild-born vs. captive-born animals in captivity?

Response: Please see our response to the comment above on this topic. Note also, that delineating the reasons for excluding papers is a recommendation of the PRISMA guidelines (L341), hence our decision to categorize papers that do not fit our search criteria.

- Line 407-408: "Of these papers, 18 (46%) were coded by 2 people for consistency." → not sure if we're missing something, but why only 18 of the papers? Did the rest not need to be coded, or just not by 2 people?

Response: Clarified, "*All 39 papers were coded by one researcher, with 18 of these (46%) also coded by a second person to ensure agreement with the coding strategy*" (L441).

- SOM: Exclusion of Clubb et al.'s elephant work for poor reporting: in fact this work did not even attempt to look at birth origin effects on reproduction, because sample sizes in the CB group were too small (very few CB elephants in zoos had reproduced at the time of the study).

Response: We understand that the primary aim of many of the papers we examined was not to compare captive-born and wild-born animals, but our systematic literature searches did detect many studies that included the necessary information that allowed their inclusion in our meta-analysis. In considering the reanalysis of our data with multiple imputation (see response to Reviewer #1), we re-examined all of the studies that we had previously excluded due to "missing data", and developed a more nuanced classification of missing data. For example, Clubb et al. reported juvenile mortality for captive-born and wild-born elephants in the Supplementary Materials, however although details are "missing" with respect to the data we require for the meta-analysis, we recognize presentation of this comparison was not the primary aim of the study, and as such have highlighted this in the caption of Supplementary Table 4.

As a final note, we should add that we very much look forward to seeing this great work published!

Georgia Mason and three graduate students (Aileen McLennan, Sam Decker and Miranda Bandeli)

Reviewer #3 (Remarks to the Author):

Review of "A meta-analysis of productivity differences in phylogenetically diverse captive environments." In this manuscript, the authors find that, across all captive breeding programs wild-origin broodstock have higher productivity in captive environments than do hatchery-origin broodstock. A number of comments follow that will hopefully be helpful in revising the manuscript.

Comments:

1. It is unclear what is gained by combining analyses across so many different types of captive breeding programs that have entirely different aims. The goals associated with captive breeding programs for producing organisms for research (e.g., zebrafish) are entirely different than the goals captive breeding programs that focus on conservation. If productivity is lower in a research program, but not a conservation program, is that meaningful? Furthermore, the definition of "productivity" is a little unclear. Sometimes it means reproductive success and other times it means traits are (assumed to be) correlated with reproductive success.

Response: Our aim was to investigate differences in reproductive success between captive-born and wild-born animals in captivity to examine general trends relevant to captive breeding as a whole. It is conventional in the presentation of meta-analysis results to provide the overall summary statistics from the meta-analysis, to determine whether a general pattern exists in the literature and whether a large degree of heterogeneity is present, indicating variation among studies. A high degree of heterogeneity in our analysis could result from variation at the study level (e.g. studies conducted in different environments) (L112). The results show that a high degree of heterogeneity in our dataset is indeed attributable to study-level processes (Supplementary Table 3). If we had found no heterogeneity in the initial stages of our analysis, we would have had no basis for examining the consequences of environmental variation among studies, and would have interpreted our main effect as the overall pattern that applied regardless of environment. Thus, our overall analyses provide the rationale for the subsequent analyses of the sources of variation, i.e. the effects of different environments.

As we have noted in our responses to the other reviewers, we have now replaced the term "productivity" with a more carefully defined term "reproductive success" (defined at L81). Some

traits are clearly correlated with reproductive success, such as fertility rate and litter size. Other traits that have a less clearly defined relationship with reproductive success, such as egg diameter, were only included if the authors of the paper discussed their relevance to reproductive success, e.g. in Aourir et al. (2013), egg volume is compared and discussed in terms of its importance to hatchling development, growth, survival and recruitment. Thus, all the traits we include are expected to be positively or negatively correlated with reproductive success (with sign adjusted as necessary to ensure all effects are in the same direction); our new definitions are intended to make this clearer in the manuscript.

2. As the authors point out, it cannot be determined if the reductions in productivity (e.g., reproductive success") are due to genetic effects of environmental effects. If the effects are entirely environmental, then the implications of these findings are perhaps overstated because a simple change in the captive program (e.g., diet) could greatly increase productivity.

Response: Our findings of decreased reproductive success of captive-born animals in captivity relative to their wild-born counterparts are of importance to a wide variety of captive industries regardless of the mechanisms resulting in this reduction. It is probable that small management changes may correct some of the differences we observe, and by providing the first systematic review of this topic we hope to encourage further investigation of the environmental and genetic effects that may be contributing to the patterns we report.

3. It is clear that there is a decrease in productivity for captive-born broodstock when looking at the log odds ratio, but it is unclear if it is biologically meaningful. How do these results translate to reproductive success? If captive-born broodstock produce 0.5% fewer offspring than wild-born offspring is that meaningful?

Response: Our systematic review does not focus on a single measure of reproductive success, as we aimed to broadly investigate differences across a range of species (where no single measure is necessarily comparable or reported). For ease of interpretation, we have backtransformed the log odds ratio estimates and provided the percentage increase or decrease in odds of reproductive success (Table 1). In the methods, we cite established benchmarks for "small", "medium" and "large" effects, qualified with the literature, which can be used to interpret odds ratio results. We have now added these categorizations throughout the manuscript when discussing results (e.g. L105, L124).

Minor comments:

Line 20-22: It is unclear how these results translate into a larger issue. What do these findings really mean?

Response: Our finding that captive-born animals were less reproductively successful across various captive environments and life-history stages is of relevance to a variety of captive managers. See above for interpretation of the magnitude of the effect sizes.

Line 24: And elsewhere, the definition of productivity is not always consistent.

Response: As indicated above, we have revised the manuscript for consistency in our terminology.

Line 62: Maternal effects can have a genetic basis.

Response: We have changed this sentence to "*Differences in reproductive success as a result of birth origin may arise as a result of genetic effects such as inbreeding depression and adaptation to captivity; non-genetic effects such as inappropriate social development, stress and nutrition; and complex interactions such as the early rearing environment and maternal effects.*" (L63-66).

Lines 63-66: I could not follow what is trying to be conveyed in this sentence.

Response: We discuss why examining multiple measures of reproductive success rather than singular measures is useful in accounting for life-history trade-offs that may occur. We have now clarified this statement and added an example: "*Due to this complexity, assessing the success of captive breeding programs by examining only one metric, such as breeding success (i.e. producing an offspring), fails to account for life-history trade-offs that may occur, and/or differential impacts of captivity throughout a species' life history. For example, if captive-born animals produce more offspring per breeding event than their wild-born counterparts, but have higher juvenile mortality,*

lifetime reproductive success (i.e. total genetic contribution to the next generation), may be similar to, or perhaps even lower, than wild-born individuals" (L66-72).

Line 77: All captive breeding programs do not aim to maximize productivity!

Response: Changed to "*As all captive breeding programs (aquaculture, conservation, laboratory research) require successful reproduction for their management objectives, all are included in this review" (L87).*

Line 90: It is unclear to me what purpose Figure 1 serves.

Response: Figure 1 shows the phylogenetic relationships among the 44 species included in our meta-analysis, and displays the depth and breadth of the species we include. The topology of this tree was a direct input to our phylogenetic meta-analysis, used to assess phylogenetic non-independence of the results. If the editor feels this Figure is unnecessary we would be willing to move it to the Supplementary.

Line 106: It would be useful to explain up front why the number of comparisons is >> than the number of studies.

Response: We have added "*some papers compared more than one reproductive measure, or for more than one species" where we first report the number of papers (39) and the number of comparisons (115), (L99).*

Line 148: Why do you believe this?

Response: We have altered "*We do not believe that our results are influenced by publication bias (Supplementary Text, Supplementary Fig. 1)" to "We found no strong evidence that our results are influenced by publication bias (Supplementary Text, Supplementary Fig. 1)" (L166).*

Could results be inflated by single outlier studies?

Response: We examined the effects of outliers and concluded that they did not change our findings. This analysis is reported in the Supplementary Text.

Line 205: Maybe. It depends on the environment and the context - there is a large body of literature on maladaptation.

Response: Changed to "*Relaxed selective pressures in captivity mean that even F1 animals that would be 'unfit' in the wild may survive to reproductive age" (L228).*

Lines 210-225: Much of this text could be omitted or re-written to be clearer.

Response: We have reduced this paragraph and rearranged for clarity (L228-244).

Lines 317-321: I think you may have missed a number of studies using only those terms.

Response: We followed PRISMA guidelines for conducting and reporting a comprehensive systematic review. It is likely that there may be some additional information in grey literature (unpublished, e.g. annual reports), but our search of published literature was comprehensive and allows us to draw useful conclusions within our clearly defined scope (L83). For example, our tests of publication bias do not indicate that our results are likely to be biased by unpublished data.

Line 346: What do you mean by "data coding strategy"?

Response: There was various information we had to extract from each paper in the systematic review, such as the species, reproductive trait and statistics. A data coding strategy refers to how we categorized this information consistently, as per PRISMA guidelines for conducting a systematic review. We have clarified this in the manuscript (L374).

REVIEWERS' COMMENTS:

Reviewer #1 (Remarks to the Author):

I have read the new manuscript and the reply letter. I think the authors did a very good job of addressing my comments and others. This will make a very nice contribution and will be widely read and cited.

Reviewer #2 (Remarks to the Author):

The authors have addressed all our comments very well, and the paper is a pleasure to read. Overall the work is extremely well conducted and fascinating in its conclusions. The one suggestion I'd make is re. line 302: sadly studbooks are not "publically" available, they are just "available" (you have to ask for them, and sometimes they say no!).

Reviewer #4 [commenting in stead of Reviewer #3] (Remarks to the Author):

I believe the authors have adequately responded to reviewer #3's comments. I thought it was an interesting piece of work, well analyzed and well written. One minor dis-satisfaction was that they don't really elaborate much on the likely causes of the counter-intuitive result that wild-born animals do better in captive environments (one would expect no difference or the opposite owing to adaptation to captivity). The only explanations I can see are drift/inbreeding, purely environmental effects of the captive environment, or in the case of aquaculture, deliberate selection for particular traits that show negative pleiotropy with fitness-related traits in captivity.

One minor point on a similar topic: lines 48-49. "Evolutionary change in captive populations is therefore likely and potentially even unavoidable as relaxation of selective pressures in captive environments leads to adaptation to captivity". They seem to confuse relaxed natural selection in captivity with domestication selection (selection for particular traits that are favored in captivity). Relaxed natural selection means genetic variation that would be disfavored in nature is neutral in captivity, leading to the accumulation of genetic variation that only becomes disadvantageous once that population returns to the wild. This is not the same as adaptation to captivity. If what they mean is the effects of small population size (drift) are overwhelming purifying selection in captivity, then that is a different issue. That results in fitness loss in captivity, not adaptation to captivity.

Manuscript NCOMMS-17-19589A, Farquharson et al., response to reviewer comments

Reviewer #1 (Remarks to the Author):

I have read the new manuscript and the reply letter. I think the authors did a very good job of addressing my comments and others. This will make a very nice contribution and will be widely read and cited.

Author response: We thank the reviewer for their comment; no edits to manuscript required

Reviewer #2 (Remarks to the Author):

The authors have addressed all our comments very well, and the paper is a pleasure to read. Overall the work is extremely well conducted and fascinating in its conclusions. The one suggestion I'd make is re. line 302: sadly studbooks are not "publically" available, they are just "available" (you have to ask for them, and sometimes they say no!).

Author response: We thank the reviewer for their comments; sentence updated as suggested.

Reviewer #4 [commenting instead of Reviewer #3] (Remarks to the Author):

I believe the authors have adequately responded to reviewer #3's comments. I thought it was an interesting piece of work, well analyzed and well written. One minor dis-satisfaction was that they don't really elaborate much on the likely causes of the counter-intuitive result that wild-born animals do better in captive environments (one would expect no difference or the opposite owing to adaptation to captivity). The only explanations I can see are drift/inbreeding, purely environmental effects of the captive environment, or in the case of aquaculture, deliberate selection for particular traits that show negative pleiotropy with fitness-related traits in captivity.

Author response: We appreciate the author's suggestions. In the submitted version, we outlined a number of possible causes of our unexpected finding that captive-born animals do worse in captivity than wild-born animals, but acknowledge that these interpretations were not clearly organised. To address this issue, we have made minor adjustments to structure in the Discussion (such as improving paragraph topic sentences), and noted the additional explanations for our results suggested by the reviewer (e.g. L233, 245 and 258 of the tracked-changes version). We feel that, as a result of these edits, the Discussion now reads much more effectively.

One minor point on a similar topic: lines 48-49. "Evolutionary change in captive populations is therefore likely and potentially even unavoidable as relaxation of selective pressures in captive environments leads to adaptation to captivity". They seem to confuse relaxed natural

selection in captivity with domestication selection (selection for particular traits that are favored in captivity). Relaxed natural selection means genetic variation that would be disfavored in nature is neutral in captivity, leading to the accumulation of genetic variation that only becomes disadvantageous once that population returns to the wild. This is not the same as adaptation to captivity. If what they mean is the effects of small population size (drift) are overwhelming purifying selection in captivity, then that is a different issue. That results in fitness loss in captivity, not adaptation to captivity.

Author response: We accept that the wording around relaxation of selection was ambiguous, and have removed that text. In its place, we have added a sentence to the Introduction that acknowledges multiple processes can lead to genetic change in captive populations relative to the wild: “Genetic change in captive populations is likely and potentially unavoidable as a result of founder effects, inbreeding, drift and adaptation to captivity, among other processes” (Line 46-48 of the track-changes version).